# Reduced-Impact Logging Maintain High Moss Diversity in Temperate Forests

Enrique Hernández-Rodríguez [1,2,*], Luis H. Escalera-Vázquez [3], Deneb García-Ávila [3], Miguel Montoro Girona [2,4,5] and Eduardo Mendoza [1,*]

[1] Instituto de Investigaciones Sobre los Recursos Naturales, Universidad Michoacana de San Nicolás de Hidalgo, Av. San Juanito Itzícuaro s/n, Morelia 58337, Michoacán, Mexico

[2] Institut de Recherche sur les Forêts, Université du Québec en Abitibi Témiscamingue, 445 Boul. de l'Université, Rouyn-Noranda, QC J9X5E4, Canada; miguel.montoro@uqat.ca

[3] Facultad de Biología, Universidad Michoacana de San Nicolás de Hidalgo, Av. Gral. Francisco J. Múgica s/n A-1, Morelia 58030, Michoacán, Mexico; lhescalera@gmail.com (L.H.E.-V.); deneb.garcia@umich.mx (D.G.-Á.)

[4] Groupe de Recherche en Écologie de la MRC-Abitibi (GREMA), Campus of Amos, Université du Québec en Abitibi Témiscamingue, 341 Rue Principal Nord, Amos, QC J9T2L8, Canada

[5] Restoration Ecology Group, Department of Wildlife, Fish and Environmental Studies, Swedish University of Agricultural Sciences, SE-901 83 Umeå, Sweden

[*] Correspondence: enrique.hernandezrodriguez@uqat.ca (E.H.-R.); eduardo.mendoza@umich.mx (E.M.)

**Abstract:** Forestry harvesting represents an important economic activity around the world. Habitat degradation due to forest harvesting contributes to biodiversity loss; therefore, it is necessary to implement logging management aimed at reducing its impact. Forest management by reduce-impact logging (RIL) involves cutting trees following regulations focused on diminishing the impact on biodiversity by following harvesting plans based on forestry inventories and participation of trained workers. In Mexico, RIL is applied mainly in temperate habitats and its effectiveness has been assessed based on vascular plants. In this study, we analyzed the diversity and community structure of terrestrial and epiphytic mosses in managed (sites number = 3) and conserved (sites number = 3) sites in the temperate forest of Sierra Juárez, Oaxaca, Mexico. Likewise, we evaluated the potential function of mosses as indicators of habitat degradation. Environmental variables were also quantified at local (canopy coverage, altitude, daily temperature, and light) and regional (total annual rainfall, orientation, and slope) scales to evaluate potential relationships with the community and species diversity. We documented 70 mosses species with a diversity (alfa, beta) and community structure similar between managed and conserved sites. For terrestrial mosses, we found marginal differences in their communities, likely related to species coverture variation in managed sites. The diversity and community structure epiphytic mosses were not statistically different in managed and conserved sites. Only the daily variation in light intensity was positively related to the variation of alpha diversity of epiphytic mosses. The species *Dicranum sumichrastii* Duby and *Leptodontium viticulosoides* (P. Beauv.) Wijk & Margad. can be considered as ecological indicators for conserved and managed sites, respectively, likely due to their relationship with light and humidity conditions. Our results suggest that that forest management by RIL could be considered as a promising tool to balance timber production and moss diversity.

**Keywords:** biodiversity conservation; bryophytes; community forest management; ecological indicator; pine-oak forest; restoration; silviculture; sustainable forest management

## 1. Introduction

Biodiversity is highly threatened by anthropogenic disturbances such as deforestation, land change, soil pollution, and habitat fragmentation around the world [1–5]. Forestry logging is a human activity that affects biodiversity patterns, specially reducing the number of species depending of the disturbance intensity [6,7]. To reduce logging impacts, sustainable

forest management represents a tool to develop productive practices and conservation including the protection of species and ecosystems services [8–10].

Forest management represents a series of techniques that enable productive practices and depending on the wood harvest intensity and the impact on species richness, six standard silvicultural systems globally can be distinguished, from the least to the most harmful: selection and retention systems, reduced-impact logging, selective logging, clear-cutting, and timber plantations [11].

Reduced-impact logging (RIL) is a silvicultural practice characterized by the selective extraction of trees from previously defined areas to reduce forest damage and protecting of riparian vegetation [12]. Trained workers select and extract individual trees carefully, according to a harvest plan and guidelines reducing the negative impact of tree felling and hauling on the remaining forest [13–17]. RIL has been implemented in the forests of Europe and Asia, showing that silviculture systems reduce impact on local and regional species richness [11,18,19]. In southern Mexico, this RIL management is developed by local communities in tropical and temperate forests [20–23]. In contrast with forest management strategies such as clear-cutting, where entire tree stands are removed and biodiversity is highly impacted [11,24], the RIL generates small clearings reducing habitat disturbance effects [25–27].

Evidence indicates that RIL has a low impact on the species richness of birds, insects, mammals, and vascular plants in tropical regions [28]. However, studies exploring the effects of RIL on native vegetation are focused on vascular plants (e.g., [29–31]), whereas effects on non-vascular flora, like mosses have not been assessed. Mosses are associated to specific microenvironmental conditions (e.g., low light, high humidity), structure of the habitat (e.g., host tree species), and type of substrates to establish and thrive (e.g., advanced decaying wood, rocks, soil) [32,33]. These micro conditions are related to richness (alpha diversity) and species turnover (beta diversity) of mosses [34]. Thus, even slight changes in environmental conditions due to RIL could negatively affect the diversity and composition of mosses communities [11,19,35]. Due to this characteristic, mosses can function as indicators of the state of forest where RIL is applied [36]. The conservation of mosses is essential because they play an important role in biochemical cycles and ecosystem dynamics due to the fact they store up to 1400% of their dry weight in water [37], fix up to 400 mg N m$^{-2}$ year$^{-1}$ [38], and provide shelter to protozoa, invertebrates, and arthropods [39]. Thus, evaluating the impact of RIL on mosses is a crucial issue to ensure biodiversity preservation in the long-term.

In this study, we evaluated the impact of RIL on moss diversity (terrestrial and epiphytic) in a pine-oak temperate forest in the state of Oaxaca, Mexico. We compared the diversity of mosses from three sites under RIL management with that of three conserved sites. Our hypotheses were: (i) the disturbance caused by RIL will affect the diversity (alpha and beta) of both groups and their community structure due to changes in microhabitat conditions (high light and temperature), and (ii) some moss species will show habitat preferences related to RIL and conserved forest sites.

## 2. Materials and Methods

### 2.1. Study Area

Our study area is located in the pine-oak forest of the Sierra Juárez (Sierra Madre) of Oaxaca state, south of Mexico [40] (Figure 1). The climate is temperate sub-humid with rains during the summer; the annual rainfall ranges between 1000 and 2000 mm, and the average annual temperature is 16 °C [41]. Pine-oak vegetation occurs in an altitudinal range from 2000 to 2800 m on humic acrisol soils [42,43]. Forest stands have species of pines (*Pinus ayacahuite* C. Ehrenb. ex Schltdl., *P. leiophylla* Schiede ex Schltdl. & Cham., *P. patula* Schltdl. & Cham., *P. pseudostrobus* Brongn.), oaks (*Quercus crassifolia* Bonpl., *Q. elliptica* Née, *Q. laeta* Liebm.), and bushes such as *Arbutus xalapensis* Kunth and *Clethra* L. spp. The most abundant genus is *Pinus* followed by *Quercus* L. [44] and present high associations with mosses like *Brachythecium* Schimp., *Bryum* Hedw., *Entodon* Müll. Hal.,

and *Leptodontium* (Müll. Hal.) Hampe ex Lindb. spp. [45,46]. In the forest understory, occur shrubs species including *Arctostaphylos pungens* Kunth, *Gaultheria acuminata* Schltdl. & Cham., *Litsea glaucescens* Kunth and herbaceous species including *Alchemilla pectinata* Kunth, *Stevia lucida* Lag., and *Dryopteris* Adans. spp. Moreover, epiphytes are represented by orchids, bromeliads, and mosses including species of the genus *Macromitrium* Brid., *Meteorium* Dozy & Molk., *Neckera* Hedw. and *Orthotrichum* Hedw. [44,46]. The area harbors a high bird diversity and threatened mammals species, and constitutes a center of vascular plant endemism in the country [40,47–49]. Approximately 74,240 ha of the Sierra Juárez are under management for timber production, almost all pine or pine-oak forest [50].

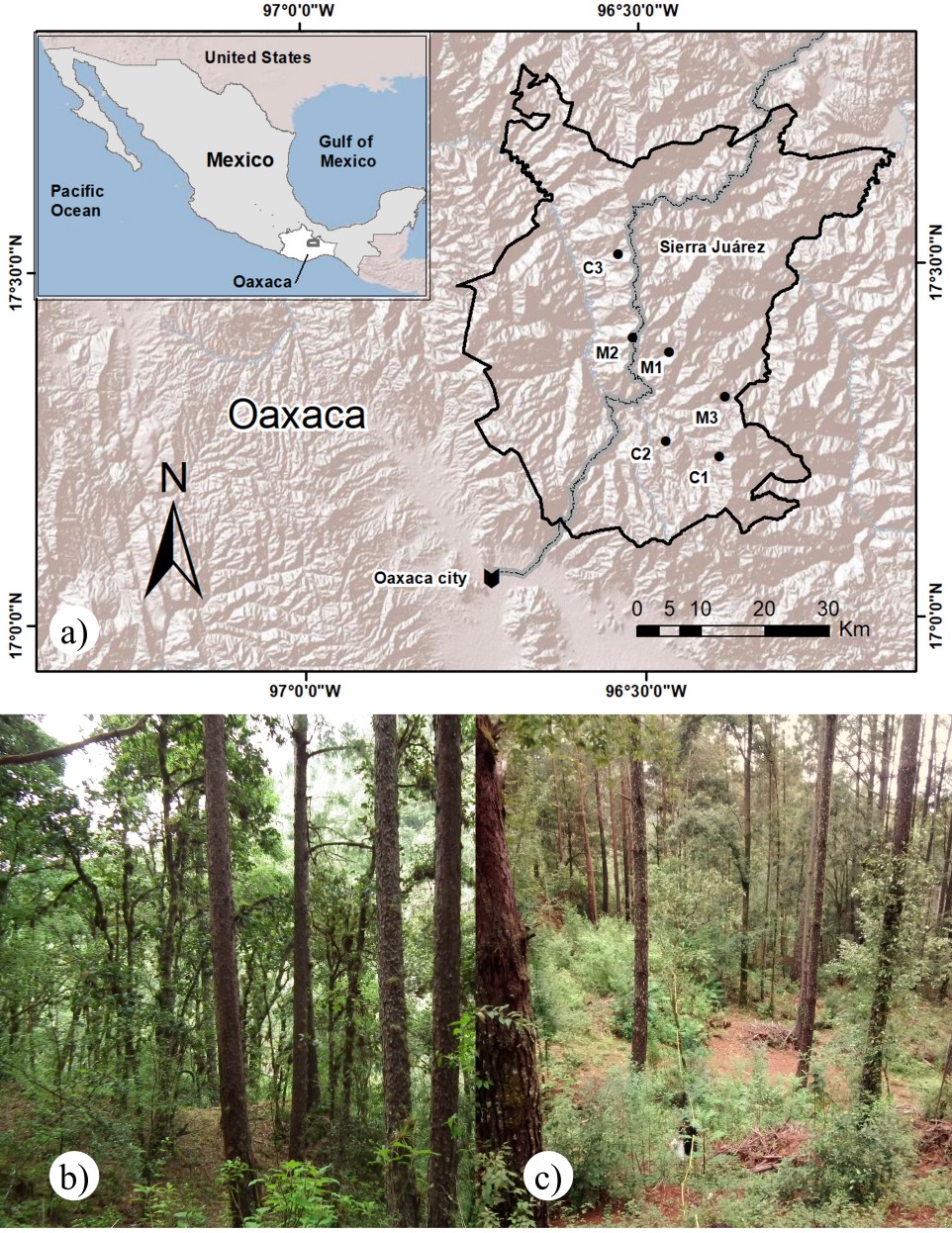

**Figure 1.** (**a**) Study sites in Sierra Juárez, Oaxaca (black circles). Conserved sites: Santa María Yavesía (C1), Santa Catarina Lachatao (C2), and San Juan Luvina (C3). Managed sites: Ixtlán de Juárez (M1), San Juan Evangelista Analco (M2), and Capulálpam de Méndez (M3). Stitch line is the road 175 Oaxaca-Tuxtepec. (**b**) Examples of conservation (C1) and (**c**) management (M2) sites.

## 2.2. Experimental Design

This study was conducted in six sites, three old forest dedicated to conservation (C) in which no logging activities are performed (untreated control), and three managed (M) where the RIL system is applied. The conserved sites differed in size C1 = 7134 ha, C2 = 12,936 ha, C3 = 230 ha. Due to the forest planning, there was variation in the time RIL management was implemented in each site. In M1, logging activities began in 2004 on 107 ha, and about in one year every ten years, diseased and old trees are removed (logging intensity). In M2, RIL started in 2012 on 13.5 ha, and in M3 in 1995 on 10 ha, in both cases, logging is conducted during periods of 10 years every 50 years in which workers prune trees branches and shrubs to reduced plant competition, favoring the increment in diameter and height of pine trees. Time since the last disturbance in each managed site is two years for M1, four for M2, and nine for M3. The tree number removed per hectare in each site during logging period is under five percentage from total trees. Additionally, RIL in the region involves forest inventories to define how the logging procedures are conducted by local trained workers and the preparation of harvest plans to minimizing the impact on vegetation [51]. Total area from these three RIL sites represented around the 0.14% from the pine-oak area in the Sierra Juarez [52].

## 2.3. Sampling and Data Compilation

Sampling was conducted during July and August 2016. In each study site, ten sampling quadrats of 1 $m^2$ were randomly established in an area of 5 ha (60 quadrats in total). In each quadrat, terrestrial mosses were those growing in the ground on bare soil, humus, and decaying wood; epiphytic mosses included those on the trunk of the closest tree (diameter at breast height $\geq$ 30 cm) of the genus *Quercus* to each of the quadrats. To sample epiphytic mosses, we place a 20 $cm^2$ quadrat on the tree trunk at the height of 1.6 m [45]. The location of the quadrat (N, S, E, or O) was set randomly [45].

To estimate the coverture of the different species of terrestrial and epiphytic, mosses quadrats of 1 and 20 $cm^2$ were divided using a grid of 100 units. We draw the area's silhouette covered by each species on a plastic sheet placed over the quadrat. The area covered by each species was measured as the percentage of units occupied in the grid. A sample of each moss was collected for taxonomic determination [45,53]. Species were identified using different taxonomical guides [54–60] and following the nomenclature in the Catalogue of Neotropical Mosses (LATMOSS) [61]. All the collected specimens were deposited in the Herbario de la Escuela de Biología of the Universidad Michoacana de San Nicolás de Hidalgo (EBUM) (Appendix A Table A1).

To assess the relationship between environmental variables with moss diversity, local-scale variables (LSV) and regional-scale variables (RSV) were measured. The LSV included canopy coverage and altitude of each quadrat, as well as daily temperature and light. The coverage of the canopy was measured in the center of each 1 $m^2$ quadrat using a convex spherical densimeter and averaging the percentages of coverage recorded when facing each cardinal point. The temperature and light were recorded in intervals of 15 min during 44 days from 3 March to 17 April 2017 (96 records per day), at each site using two automatic recorders (HOBOS UA-002-64, Onset Computer Corporation, MA, USA). The two recorders were placed in the center of each 5 ha study site separated by 223 m. The altitude was registered with a handheld GPS (GPSMAP 64s, GARMIN, KN, USA). RSV variables included total annual rainfall, orientation, and slope of each site. These were obtained from the Instituto Nacional de Estadística y Geografía (INEGI) databases with a 30 m of resolution [62] and processed using Quantum Geographic Information System (QGIS, Open Source Geospatial Foundation Project) [63]. We obtained the information overlapping the INEGI environmental shapefiles with each site location.

## 2.4. Data Analyses

To assess sampling completeness in each site, we calculated the Chao 2 species richness index (alfa diversity) [64]. To compare observed and estimated richness, rarefac-

tion/extrapolation (R/E) curves were performed with 95% confidence intervals. We calculated the percentage that the observed species represented in the estimated richness. To assess alpha diversity and perform the R/E curves, the method proposed by Chao [64] based on the Hill number of zero-order (q = 0) was used. Beta diversity was estimated to evaluate how the species replacement ($\beta$repl) and species richness ($\beta$rich) affected total beta diversity ($\beta$total) [65] within and between sites.

We performed a non-metric multidimensional scaling analysis (NMDS), using Bray–Curtis dissimilarities matrices, analysis of similarities (ANOSIM test), and rank abundance curves [66] to compare the moss communities between forest types. For NMDS and ANOSIM, we used data to the quadrant level. We discarded the species occurring in less than 5% of the quadrats to focus only on those most representative and because rare species can have a high influence in community simulation tests [67–69]. The percentage of coverture was transformed (by applying the square root and arcsine) prior to analysis to have data with a normal distribution [69]. We conducted the ANOSIM test with 999 permutations. To build rank-abundance curves, we used the coverture data of each species transformed to the logarithm. To evaluate the moss species association with forest types, we applied the indicator value (IndVal index) [70] using the quadrants data per site. This method considers the specificity (A): the degree of association of a species with a site type, and fidelity (B): the degree to which, given that there is a forest conserved or under management, a certain moss species occurs. The statistical significance of each relationship was evaluated with 999 permutations [70].

Finally, to analyze the relationship between abiotic environmental variables and moss diversity (richness values, mean, standard deviation), the averages of each LSV and RSV variable at site level were calculated. Then, their relationship was analyzed through a correlation analysis using the Pearson correlation coefficient [71]. Additionally, the averages values per site were used to evaluate differences in LSV and RSV between management and conservation conditions applying t-student and Wilcoxon test.

All analyses were conducted using R (Version 3.5.0, R Foundation for Statistical Computing, Vienna, Austria) [72]. For richness estimation and rarefaction/extrapolation curves, we used iNEXT 2.0.14 package [73]; for the analysis of beta diversity, the BAT package [74]; for the NMDS and ANOSIM analyzes, we used vegan 2.4-4 package [75]; for the calculation of IndVal index, we used Indicspecies 1.7.1 package [76]; to conduct the correlation analysis, the Performance Analytics package [77].

## 3. Results

### 3.1. Alpha and Beta Diversity and Community Structure of Mosses

We obtained 251 samples, which included 70 species of 51 genera and 31 families of mosses (Appendix A Table A1). Fifteen mosses species were terrestrials and epiphytic, 45 were exclusively terrestrial and exclusively 20 epiphytic. Based on the Chao-2 species richness estimator, the sampling's completeness was, on average of 50% for terrestrial and 48% for epiphytic mosses per site, respectively. Managed sites had, on average, $12 \pm 2$ species (mean $\pm$ SD) of terrestrial mosses, while conserved sites had $18 \pm 1$ species. In comparison, there was an average of $10 \pm 3$ species and $10 \pm 1$ species epiphytic mosses in management and conservation sites (Table 1). Thus, the average of epiphytic moss richness was similar in conserved and management sites.

**Table 1.** Observed (Obs.) and estimated (Est.) richness for terrestrial and epiphytic mosses among the study sites (C = conserved; M = managed) by Chao 2 index. CI = Confidence intervals; % spp. record = percentage of species recorded.

| Site | Terrestrial | | | Epiphytic | | |
|---|---|---|---|---|---|---|
| | Obs. | Est. (CI) (Chao 2) | % spp. Record | Obs. | Est. (CI) (Chao 2) | % spp. Record |
| C1 | 17 | 39 (25–78) | 45 | 7 | 18 (10–55) | 39 |
| C2 | 19 | 28 (22–44) | 68 | 11 | 40 (18–123) | 28 |
| C3 | 17 | 49 (28–115) | 35 | 12 | 15 (13–21) | 80 |
| M1 | 10 | 13 (11–20) | 77 | 9 | 13 (10–23) | 69 |
| M2 | 11 | 40 (18–123) | 28 | 11 | 47 (21–150) | 23 |
| M3 | 14 | 28 (18–62) | 50 | 9 | 12 (10–19) | 75 |

The R/E curves showed no significant differences (based in the lack of overlap between 95% confidence intervals) between managed and conserved sites for terrestrial and epiphytic mosses. In both groups of mosses, species richness trends were maintained when extrapolating to greater sampling (Figure 2).

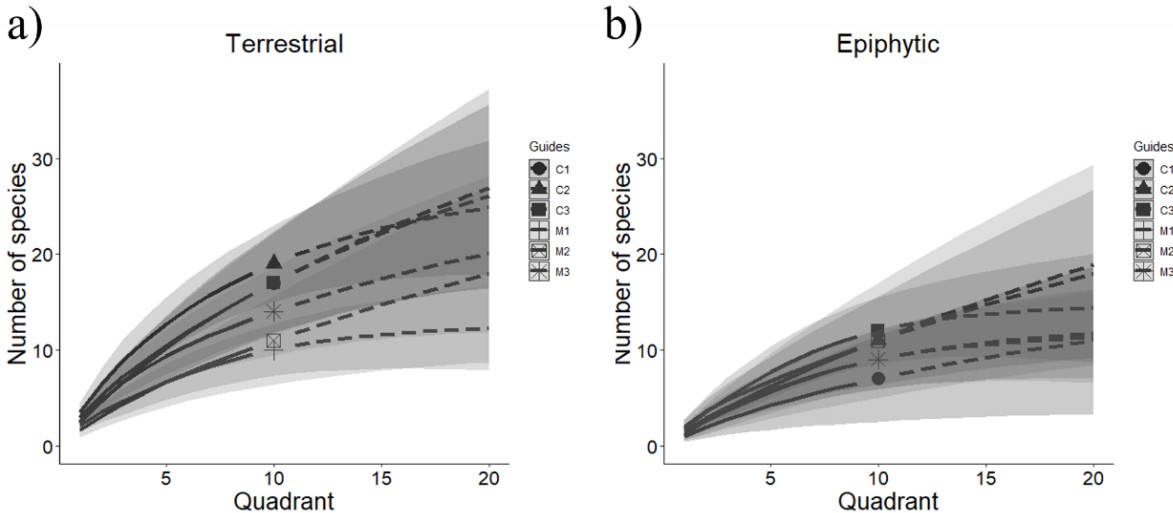

**Figure 2.** Rarefaction curves (R/E curves) with intrapolation (solid line)/extrapolation (dotted line) of the species richness of (**a**) terrestrial and (**b**) epiphytic mosses among the study sites (Guides). Each curve has confidence intervals of 95%. Confidence intervals overlapping indicate no significant differences in the species richness observed and estimated between managed and conservation sites.

Beta diversity values for terrestrial and epiphyte mosses are similar in conserved and managed sites. In conserved sites and for terrestrial mosses, we estimated a total β diversity between 71 and 80%. Between 64 and 79% of β diversity was explained by species replacement (βrepl) and only between 0 to 7% was explained by differences in richness (βrich). In comparison, within managed sites, we estimate a total β between 69 and 81%. From this, between 53 and 67% was explained by βrepl and between 6 to 21% by βrich. Thus, in both forest types, βtotal diversity for terrestrial mosses was mainly associated with βrepl (Table 2). In conserved sites for epiphytic mosses, the βtotal ranged between 62 and 85%. βrepl explained 31 to 80% of this β diversity, and βrich between 5 and 31%. In managed sites, the total β ranged between 67 and 100%, whereas βrepl represented 53 to 90%, and βrich between 10 to 13%. In summary, in both forest types, the βtotal diversity was mostly explained by βrepl, and in conserved sites, βrich was higher than in managed sites (Table 2). Between forest types, the total β diversity of terrestrial and epiphytic mosses varied between 76 to 74%, respectively. For terrestrial mosses, 52% of this β diversity was explained by βrepl, whereas the remaining 24% was explained by βrich. In the case of epiphytic mosses, βtotal was defined entirely by βrepl (Table 2).

**Table 2.** Beta diversity of terrestrial and epiphytic mosses between conserved and managed forests. Species replacement (βrepl), differences in richness (βrich), and total beta diversity (βTot) are expressed in percentage.

| | Terrestrial | | | Epiphytic | | |
|---|---|---|---|---|---|---|
| **Sites** | **βrepl** | **βrich** | **βTot** | **βrepl** | **βrich** | **βTot** |
| C1-C2 | 73 | 7 | 80 | 31 | 31 | 62 |
| C1-C3 | 79 | 0 | 79 | 50 | 31 | 81 |
| C2-C3 | 64 | 7 | 71 | 80 | 5 | 85 |
| M1-M2 | 63 | 6 | 69 | 53 | 13 | 67 |
| M1-M3 | 53 | 21 | 74 | 88 | 0 | 88 |
| M3-M2 | 67 | 14 | 81 | 90 | 10 | 100 |
| C-M | 52 | 24 | 76 | 74 | 0 | 74 |

For terrestrial and epiphytic mosses, the rank-abundance curves had at each site in both forest types a steep slope. This suggests a low uniformity in the community dominance because a few species have a high abundance. However, the most abundant species, and those with low abundance, differed between sites. For terrestrial mosses, there were 1–5 species with the lowest abundance in the managed forest, while in the conserved forest, there are 6–7 species. There were a few dominant terrestrial moss species, most of them with high abundance, while in epiphytic mosses, there are more dominant species, but few of them had low abundance (Figure 3). Common terrestrial species in both forest types included *Braunia squarrulosa* (Hampe) Müll. Hal., *Bryum billarderii* Schwägr., *Hypnum amabile* (Mitt.) Hampe, *Mittenothamnium reptans* (Hedw.) Cardot, *Sematophyllum swartzii* (Schwägr.) W.H. Welch & H.A. Crum, *Thuidium delicatulum* (Hedw.) Schimp., *Trichostomum brachydontium* Bruch, and *Zygodon ehrenbergiI* Müll. Hal. In epiphytic mosses, common species were *Holomitrium pulchellum* Mitt., *Leptodontium viticulosoides* (P. Beauv.) Wijk & Margad., *Leucodon curvirostris* Hampe, *Neckera chlorocaulis* Müll. Hal., and *Zygodon viridissimus* (Dicks.) Brid.

The NMDS ordination procedure suggests that the community of terrestrial and epiphytic mosses are similar between managed and conserved sites (Figure 4). However, for the terrestrial moss community, there are higher R values and significant differences (ANOSIM $R = 0.14$; $p = 0.001$) in comparison to the epiphytic moss community (ANOSIM $R = 0.02$; $p = 0.199$). Differences in terrestrial moss community can be related to higher variation in species coverture in some managed sites. For epiphytic mosses, some quadrants in management sites are distant from the rest due to differences in species composition but without significant differences (Figure 4). The low-stress values in both cases indicate stability in the ordination configuration [78].

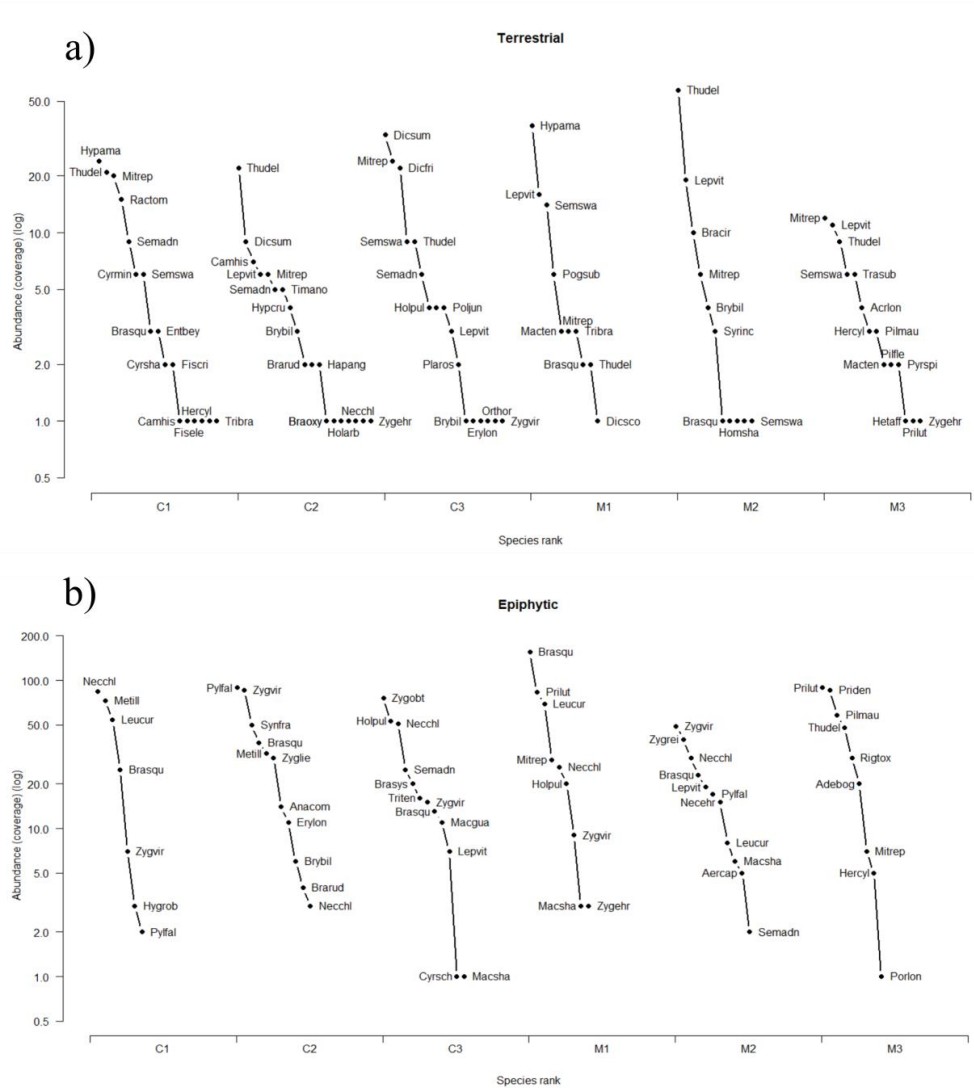

**Figure 3.** Rank-abundance curves of terrestrial (**a**) and epiphytic (**b**) mosses for each site, in managed and conservation forests. The acronyms correspond to the names of the species in Appendix A Table A1.

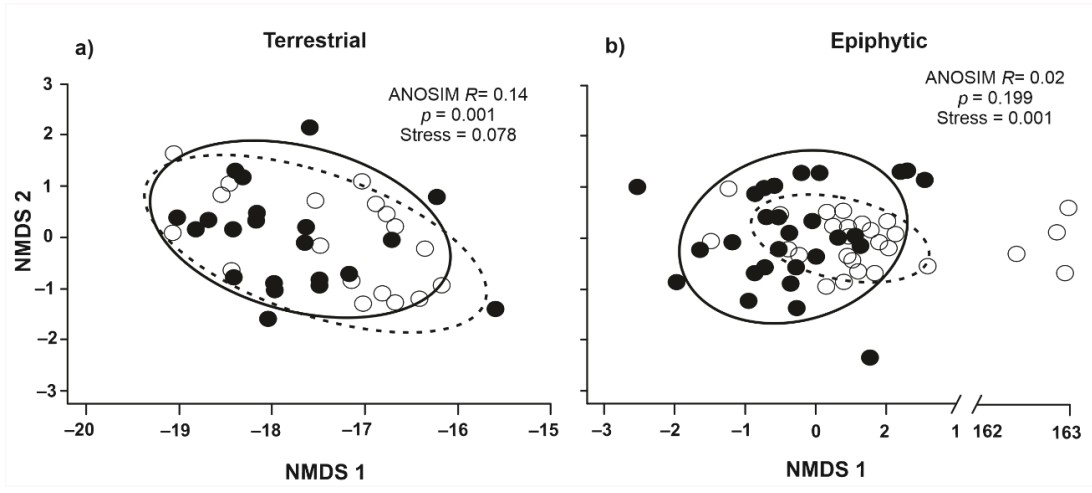

**Figure 4.** Non-metric multidimensional scaling (NMDS) of the terrestrial (**a**) and epiphytic (**b**) mosses in managed sites (hollow black circles and dotted ellipse) and conserved (solid black circles and continuous ellipse).

*3.2. Indicator Species and the Environment Influence on Mosses Diversity*

The IndVal index identified the terrestrial mosses *Dicranum sumichrasti* and *Leptodontium viticulosoides* as potential indicator species for conservation and management sites, respectively ($p < 0.05$). *D. sumichrasti* is an indicator of sites conserved with high specificity (A = 100%) but low fidelity (B = 30%). In contrast, *L. viticulosoides* is an indicator of sites under management with high specificity (A = 84%) but intermediate fidelity (B = 53%). In the case of the epiphytic mosses, no indicator species were found.

We did not find significant differences in local-scale variables (LSV) and regional-scale variables (RSV) between managed and conserved sites (Table 3). Of the LSV and RSV analyzed by correlation analysis, RSV did not show an effect on mosses diversity. Nevertheless, we found a positive correlation between the variation in alpha diversity of epiphytic mosses and the variation in light availability (LSV) ($r = 0.83$, $p < 0.05$) (Appendix A Table A2).

**Table 3.** Local and regional variables comparison between managed and conserved sites using T-student or Wilcoxon test.

|  | **Variables** | *t* **or** *Wilcoxon* **Value** | *p* |
|---|---|---|---|
| Local-scale variables (LSV) | Canopy coverage (%) | W = 7 | 0.35 |
|  | Altitude (m a.s.l) | $t = -1.5$ | 0.22 |
|  | Daily temperature (°C) | $t = 2.6$ | 0.07 |
|  | Daily light (lx) | $t = -0.5$ | 0.66 |
| Regional-scale variables (RSV) | Total annual rainfall (mm) | $t = -1.8$ | 0.16 |
|  | Orientation (°) | W = 7 | 0.40 |
|  | Slope (°) | $t = 1.4$ | 0.23 |

## 4. Discussion

The evaluation of silvicultural practices to reach sustainable forest management strategies is a major priority in forest ecosystems around the world [79–83]. We are living a critical moment in the adaptation of harvesting methods to maintain the forest biodiversity [84–89]. Here, we present the first study evaluating the effect of forest management by RIL on moss diversity in a temperate forest in Mexico. Based on our results, the RIL can keep a great diversity (alpha and beta), but also the community structure, of this group of plants in managed patches.

*4.1. Diversity and Community Moss Structure*

Our study demonstrated that RIL helps to conserve terrestrial and epiphytic mosses diversity in temperate forest. These results contrast with studies showing a decrease in moss diversity as a consequence of clear-cutting [90,91] and selective logging [92]. Our results also differ from those where other management systems such as shelterwood logging [93], clear-cutting [94], or tree retention [95] are applied and species richness is found to be similar between managed and protected sites but where species composition differs due to changes in the forest structure. These differences are related with disturbance severity of each management system. Compared with those system RIL harvesting areas of Sierra Juárez, they do not show severe impacts on the diversity and moss community structure base on their similarity with conserved sites. However, we found a marginal effect on the terrestrial moss community (Figure 4). In our study, the richness and community structure of terrestrial mosses are similar (Figures 2–4; Table 2), but our ANOSIM analysis reveals differences, possibly related to species coverture variation in RIL management sites. This finding highlights the importance of including other bryophyte community attributes, like their coverture, to know disturbance effects that could be overlooked.

The similarity in the richness and community moss structure between conserved and managed forests is likely related to maintaining microhabitat conditions [96–98]. These micro conditions are, for example, the humidity and bark substrate provided by *Quercus* trees keeping and the no severe disturbance on terrestrial substrates in RIL areas [99]. In this way, general characteristics of the structure of moss communities remained similar for both groups in terms of the frequency and composition of dominant and rare species.

Based on our results, RIL does not affect the community level, but it could have effects on some particular species. This possibility is reinforced if we consider that the inventory completeness (alpha diversity) at each site was approximately 50%. In the case of terrestrial mosses in the three sites, rank-abundance curves indicated that RIL could affect some of the less abundant species (or rare local species [100]) because of their specific requirements [101] compared to the most abundant ones (Figure 3a). These requirements are represented, for example, by availability of suitable habitats as deadwood with different decay stages, humidity, and special edaphic conditions [102,103]. Another factor in explaining these results is the time since the last disturbance. For example, site M1 with a recent disturbance (two years) had fewer species observed and expected than other managed sites (Table 1). Some of these terrestrial species do not present in managed sites including *Dicranum scoparium* Hedw., *Heterophyllum affine* (Hook. in Kunth) Fleisch., *Prionodon luteovirens* (Taylor) Mitt., and *Zygodon ehrenbergii* Müll. Hal. Among epiphytes, the M3 site was different from the rest of the managed and conserved sites which can be related to its stand conditions. This site is the oldest, with less logging area, and presents the highest humidity compared with the rest of the evaluated management sites. An old forest keeps conditions such as humidity and unique subtracts necessary for some species [104,105]. Other studies in boreal forests indicated that differences in the intensity disturbance could change the bryophytes species composition over time [106].

### 4.2. Indicator Species

Regarding the RIL effects on the species level, we identified *D. sumichrastii* as an indicator species for conserved sites and *L. viticulosoides* as an indicator species for sites under management. These mosses have been previously found to be associated with conserved and disturbed sites, respectively [107,108]. They have characteristics as stem size up to 6 cm and growing forms in tufts [60] that can facilitate their determination in the field and, therefore, their use as indicator species by non-specialists. However, since *L. viticulosoides* also appears as a common epiphytic moss in both forest types, it would be an adequate indicator only when occurring as terrestrial moss. At the same time, this species is common in open sites [58], which are not necessarily linked to management. In the case of *D. sumichrastii*, it is associated with shaded and partially open conditions [58], characteristics of common occurrence in conserved sites. Given its high percentage of specificity, it is an adequate indicator of this type of forest. For the epiphytic mosses, since the forest management did not significantly affect their communities, no species associated with any forest type were obtained. Thus, we recommend the use of *L. viticulosoides* and *D. sumichrastii* as ecological indicators of habitat conditions and forest quality monitoring in temperate forests of the Sierra Juarez, Oaxaca, Mexico.

### 4.3. Implications for Forest Management and Future Research

Our results supported the documented low effects of RIL on other taxa and determine that these forest practices with low levels of disturbances make it a suitable alternative to maintain high biodiversity [11,109,110]. To guaranty the biodiversity conservation of Sierra Juarez forests, it is crucial to consider the RIL harvesting areas compared to the surrounding conserved forest extensions, keeping the continuity of suitable conditions for mosses [26,111–113]. Thus, landscape features should be taken into account to explain the diversity levels in RIL harvesting areas in future studies [50,114]. Because our study is based on three sites under management by RIL, we recommend developing future research adding more study sites in order to increase the completeness of the moss inventory and obtain more fine results. In addition, to obtain a fine understanding of the effects of this silviculture strategy on biodiversity, as well as to study other taxonomical groups as fungi communities. We suggest studying other silvicultural methods in the region which could be an alternative to balance biodiversity conservation and the forest management developed by local communities (e.g., [115,116]).

Including ecologically important taxa such as mosses in the evaluation of RIL impacts will be useful to provide a more comprehensive assessment of the effects of this forest management strategy on the biodiversity which is of special relevance in areas with a high floristic diversity such as our study area.

## 5. Conclusions

This study supports the use of RIL as a forest management strategy able to maintain the diversity (alpha and beta) and structure of the moss community. This comprises 70 moss species including terrestrial and epiphytic taxa in the temperate forest in the Sierra Juarez, Oaxaca, Mexico. To conserve diversity in managed areas, it will be recommendable to develop strategies to preserve microenvironmental characteristics. For example, to continue with the conservation of *Quercus* trees in logging sites. Therefore, this research demonstrated the potential of *D. sumichrasti* and *L. viticulosoides* as indicator species of conserved and managed sites, respectively. In addition, because RIL is developed by local communities, these results support the positive outcomes in the management and conservation initiatives of their forest.

Finally, although the RIL system seems to help the conservation of a wide diversity of terrestrial and epiphytic mosses, it is recommended that further studies be developed to understand whether this type of management can also benefit, for example, the biomass and coverture of populations, as well as the development of processes such as water storage in which these plants participate. Similarly, the study of other biological groups understudied under RIL management may contribute to a better understanding of the effects of a forest management system that promises to be compatible with the conservation of biodiversity.

**Author Contributions:** Conceptualization: E.H.-R., E.M. Methodology: E.M., L.H.E.-V., E.H.-R., D.G.-Á. Samples taxonomic determination: E.H.-R., D.G.-Á. Data curation: E.H.-R., L.H.E.-V. Formal analysis: E.H.-R., E.M., L.H.E.-V. Interpretation results: E.H.-R., E.M., L.H.E.-V., M.M.G. Investigation: M.M.G., L.H.E.-V., E.H.-R. Project administration: E.H.-R., E.M. Resources: M.M.G. Supervision: E.M., M.M.G., L.H.E.-V. Validation: E.M., M.M.G., L.H.E.-V. Visualization: M.M.G., L.H.E.-V., E.H.-R. Writing—original draft: E.H.-R., E.M., L.H.E.-V., M.M.G. Writing—review & editing: E.H.-R., E.M., M.M.G., L.H.E.-V., D.G.-Á. Funding acquisition: E.H.-R., E.M. All authors have read and agreed to the published version of the manuscript.

**Funding:** This research was funded by the National Council of Science and Technology (CONACYT) (grant number 595991), The Sigma Xi Grant-in-Aid of Research (grant number G201603152071095), and PRODEP network: "Biodiversity conservation in anthropized environments" (No. Project 180790 of SEP-CONACYT).

**Acknowledgments:** We are grateful to different municipalities from Sierra Juárez for allow us to develop this work in their forests; to Bruce Allen to provide bibliographic material for the taxonomic determination of samples; to Jesus López Santiago and Violeta Sarai Jiménez Hernández for their support in the fieldwork. Also, we thank Nicole Fenton and *Corrige-moi* group of the Université du Québec en Abitibi-Témiscamingue for their valuable comments to this manuscript.

**Conflicts of Interest:** The authors declare no conflict of interest.

## Appendix A

**Table A1.** Moss species checklist in conserved and managed areas with reduce-impact logging in the Sierra Juárez, Oaxaca. Species presence in each site is expressed with a T for terrestrial, E for the epiphytic (collected-on *Quercus*), and TE for those collected in both samplings. The acronyms are used in the rank-abundance curves.

| Acronym | Families and Species | Sites | | | | | |
|---------|----------------------|-------|------|------|------|------|------|
| | | **C1** | **C2** | **C3** | **M1** | **M2** | **M3** |
| | **Amblystegiaceae** | | | | | | |
| Anacom | *Anacamptodon compactus* (Thér.) W.R. Buck | | E | | | | |
| Camhis | *Campylium hispidulum* (Brid.) Mitt. | T | T | | | | |
| Hygrob | *Hygrohypnum robinsonii* H.A. Crum | E | | | | | |
| | **Brachytheciaceae** | | | | | | |
| Aercap | *Aerolindgia capillacea* (Hornsch.) M. Menzel | | | | | E | |
| Braoxy | *Brachythecium oxycladon* (Brid.) A. Jaeger | | T | | | | |
| Bracir | *Brachythecium cirriphylloides* K.D. McFarland | | | | | T | |
| Brarud | *Brachythecium ruderale* (Brid.) W.R. Buck | | TE | | | | |
| | **Bryaceae** | | | | | | |
| Brasys | *Brachymenium systylium* (Müll. Hal.) A. Jaeger | | | E | | | |
| Brybil | *Bryum billarderii* Schwägr. | | TE | T | | T | |
| Rhohui | *Rhodobryum huillense* (Welw. &Duby) Touw | | | | | T | |
| | **Calymperaceae** | | | | | | |
| Syrinc | *Syrrhopodon incompletus* Schwägr. | | | | | T | |
| | **Daltoniaceae** | | | | | | |
| Adebog | *Adelothecium bogotense* (Hampe) Mitt. | | | | | | E |
| | **Dicranaceae** | | | | | | |
| Dic1 | Dicranaceae 1 | | | T | | | |
| Dicfrig | *Dicranum frigidum* Müll. Hal. | | | T | | | |
| Dicsco | *Dicranum scoparium* Hedw. | | | | T | | |
| Dicsum | *Dicranum sumichrastii* Duby | | T | T | | | |
| Holarb | *Holomitrium arboreum* Mitt. | | T | | | | |
| Holpul | *Holomitrium pulchellum* Mitt. | T | | TE | E | | |
| | **Entodontaceae** | | | | | | |
| Entbey | *Entodon beyrichii* (Schwägr.) Müll. Hal. | T | | | | | |
| Erylon | *Erythrodontium longisetum* (Hook.) Paris | | E | T | | | |
| | **Fissidentaceae** | | | | | | |
| Fiscri | *Fissidens crispus* Mont. | T | | | | | |
| Fisele | *Fissidens elegans* Brid. | T | | | | | |
| | **Hedwigiaceae** | | | | | | |
| Brasqu | *Braunia squarrulosa* (Hampe) Müll. Hal. | TE | TE | E | TE | TE | |

Table A1. *Cont.*

| Acronym | Families and Species | Sites | | | | | |
|---|---|---|---|---|---|---|---|
| | | C1 | C2 | C3 | M1 | M2 | M3 |
| | **Hypnaceae** | | | | | | |
| Hercyl | *Herzogiella cylindricarpa* (Cardot) Z. Iwats. | T | | | | | TE |
| Homsha | *Homomallium sharpii* Ando & Higuchi | | | | | T | |
| Hypama | *Hypnum amabile* (Mitt.) Hampe | T | | | T | | |
| Hypcup | *Hypnum cupressiforme* Hedw. | | T | | | | |
| Mitrep | *Mittenothamnium reptans* (Hedw.) Cardot | T | T | T | TE | T | TE |
| Pylfal | *Pylaisia falcata* Schimp. | E | E | | | E | |
| | **Lembophyllaceae** | | | | | | |
| Pilfle | *Pilotrichella flexilis* (Hedw.) Ångstr. | | | | | | T |
| Pilmau | *Pilotrichella mauiensis* (Sull.) A. Jaeger | | | | | | TE |
| | **Leskeaceae** | | | | | | |
| Hapang | *Haplocladium angustifolium* (Hampe&Müll. Hal.) Broth. | | T | | | | |
| | **Leucodontaceae** | | | | | | |
| Leucur | *Leucodon curvirostris* Hampe | E | | | E | E | |
| | **Meteoriaceae** | | | | | | |
| Metill | *Meteorium illecebrum* Sull. | E | E | | | | |
| | **Mniaceae** | | | | | | |
| Plaros | *Plagiomnium rostratum* (Schrad.) T.J. Kop. | | | T | | | |
| | **Neckeraceae** | | | | | | |
| Necchl | *Neckera chlorocaulis* Müll. Hal. | E | TE | TE | E | E | |
| Necehr | *Neckera ehrenbergii* Müll. Hal. | | | | | E | |
| Porlon | *Porotrichum longirostre* (Hook.) Mitt. | | | | | | E |
| | **Orthotrichaceae** | | | | | | |
| Macten | *Macrocoma tenuis* (Hook. &Trev.) Vitt | | T | | T | | T |
| Macgua | *Macromitrium guatemalense* Müll. Hal. | | | E | | | |
| Macsha | *Macromitrium sharpii* H.A. Crum ex Vitt | | | E | E | E | |
| Orthor | *Orthotrichum hortoniae* Vitt | | | T | | | |
| Zygehr | *Zygodon ehrenbergii* Müll. Hal. | | T | | E | | T |
| Zyglie | *Zygodon liebmannii* Schimp. | | E | | | | |
| Zygobt | *Zygodon obtusifolius* Hook. | | | E | | | |
| Zygrei | *Zygodon reinwardtii* (Hornsch.) A. Braun | | | | | E | |
| Zygvir | *Zygodon viridissimus* (Dicks.) Brid. | E | E | TE | E | E | |
| | **Pilotrichaceae** | | | | | | |
| Trasub | *Trachyxiphium subfalcatum* (Hampe) W.R. Buck | | | | | | T |
| | **Polytrichaceae** | | | | | | |
| Pogsub | *Pogonatum subflexuosum* (Lorentz) Broth. | | | | T | | |
| Poljun | *Polytrichum juniperinum* Hedw. | | | T | | | |

**Table A1.** *Cont.*

| Acronym | Families and Species | Sites | | | | | |
|---------|----------------------|-------|----|----|----|----|----|
| | | C1 | C2 | C3 | M1 | M2 | M3 |
| | **Pottiaceae** | | | | | | |
| Hyoinv | *Hyophila involute* (Hook.) A. Jaeger | T | | | | | |
| Lepvit | *Leptodontium viticulosoides* (P. Beauv.) Wijk & Margad. | | T | TE | T | TE | T |
| Synfra | *Syntrichia fragilis* (Taylor) Ochyra | | E | | | | |
| Timano | *Timmiella anomala* (Bruch & Schimp.) Limpr. | | T | | | | |
| Tribra | *Trichostomum brachydontium* Bruch | T | | | T | | |
| Triten | *Trichostomum tenuirostre* (Hook. & Taylor) Lindb. | | | E | | | |
| | **Prionodontaceae** | | | | | | |
| Priden | *Prionodon densus* (Sw. ex Hedw.) Müll. Hal. | | | | | | E |
| Prilut | *Prionodon luteovirens* (Taylor) Mitt. | | | | | E | TE |
| | **Pylaisiadelphaceae** | | | | | | |
| Hetaff | *Heterophyllium affine* (Hook. in Kunth) Fleisch. | | | | | | T |
| | **Racopilaceae** | | | | | | |
| Ractom | *Racopilum tomentosum* (Hedw.) Brid. | T | | | | | |
| | **Rhabdoweisiaceae** | | | | | | |
| Symvag | *Symblepharis vaginata* (Hook.) Wijk & Margad. | | T | | | | |
| | **Rhizogoniaceae** | | | | | | |
| Pyrspi | *Pyrrhobryum spiniforme* (Hedw.) Mitt. | | | | | | T |
| | **Rigodiaceae** | | | | | | |
| Rigtox | *Rigodium toxarion* (Schwägr.) A. Jaeger | | | | | | E |
| | **Sematophyllaceae** | | | | | | |
| Acrlon | *Acroporium longirostre* (Brid.) W.R. Buck | | | | | | T |
| Semadn | *Sematophyllum adnatum* (Michx.) E. Britton | T | T | TE | | TE | |
| Semswa | *Sematophyllum swartzii* (Schwägr.) W.H. Welch & H.A. Crum | T | | T | T | T | T |
| | **Thuidiaceae** | | | | | | |
| Cyrmin | *Cyrto-hypnum minutulum* (Hedw.) W.R. Buck & H.A. Crum | T | T | T | | | |
| Cyrsch | *Cyrto-hypnum schistocalyx* (Müll. Hal.) W.R. Buck & H.A. Crum | | | E | | | |
| Cyrsha | *Cyrto-hypnum sharpii* (H.A. Crum) W.R. Buck & H.A. Crum | T | | | | | |
| Thudel | *Thuidium delicatulum* (Hedw.) Schimp. | T | T | T | T | T | TE |

**Table A2.** Correlation analysis values between alpha diversity and environmental variables. TMR = Terrestrial mosses richness; EMR = Epiphytic mosses richness; M_RTM = Mean richness of terrestrial mosses; M_REM = Mean richness of epiphytic mosses; SD_TMR = Standard deviation of the terrestrial mosses richness; SD_EMR = Standard deviation of epiphytic mosses richness; M_DT = Mean daily temperature; SD_DT = Standard deviation of the daily temperature; M_DL = Mean of the daily light; SD_DL = Standard deviation of the daily light; M_CC = Mean of the canopy coverage; SD_CC = Standard deviation of the canopy coverage; M_Alt = Mean of altitude; M_TAP = Mean of the total annual precipitation; M_Ori = Mean of the orientation; M_Slop = Mean of the slope. Bold numbers indicate correlations > 0.83.

| | TMR | EMR | M_RTM | M_REM | SD_TMR | SD_EMR | M_DT | SD_DT | M_DL | SD_DL | M_CC | M_Alt | M_Tap | M_Ori | M_Slop | SD_CC |
|---|---|---|---|---|---|---|---|---|---|---|---|---|---|---|---|---|
| TMR | 1 | −0.09 | 0.8 | 0.21 | 0.21 | −0.09 | 0.59 | −0.12 | −0.38 | −0.35 | 0.57 | −0.51 | −0.62 | −0.02 | 0.61 | −0.63 |
| EMR | −0.09 | 1 | 0.07 | 0.67 | 0.12 | 0.65 | 0.15 | 0.27 | 0.64 | 0.54 | −0.7 | 0.25 | −0.37 | −0.2 | 0.11 | 0.69 |
| M_RTM | 0.8 | 0.07 | 1 | 0.63 | 0.49 | −0.28 | 0.57 | −0.53 | −0.54 | −0.57 | 0.65 | −0.47 | −0.35 | −0.13 | 0.72 | −0.67 |
| M_REM | 0.21 | 0.67 | 0.63 | 1 | 0.21 | 0.08 | 0.49 | −0.21 | 0.11 | 0 | −0.05 | −0.07 | −0.37 | 0.05 | 0.45 | 0.05 |
| SD_TMR | 0.21 | 0.12 | 0.49 | 0.21 | 1 | −0.07 | −0.11 | −0.81 | −0.58 | −0.58 | 0.34 | −0.25 | 0.44 | **−0.87** | 0.46 | −0.32 |
| SD_EMR | −0.09 | 0.65 | −0.28 | 0.08 | −0.07 | 1 | 0.32 | 0.46 | 0.77 | **0.83** | −0.69 | −0.2 | −0.36 | −0.33 | 0.25 | 0.71 |
| M_DT | 0.59 | 0.15 | 0.57 | 0.49 | −0.11 | 0.32 | 1 | 0 | 0.12 | 0.2 | 0.26 | −0.78 | −0.68 | 0.15 | **0.82** | −0.23 |
| SD_DT | −0.12 | 0.27 | −0.53 | −0.21 | −0.81 | 0.46 | 0 | 1 | 0.8 | 0.78 | −0.67 | 0.39 | −0.59 | 0.54 | −0.47 | 0.62 |
| M_DL | −0.38 | 0.64 | −0.54 | 0.11 | −0.58 | 0.77 | 0.12 | 0.8 | 1 | 0.98 | **−0.9** | 0.27 | −0.44 | 0.25 | −0.24 | **0.91** |
| SD_DL | −0.35 | 0.54 | −0.57 | 0 | −0.58 | **0.83** | 0.2 | 0.78 | 0.98 | 1 | **−0.84** | 0.11 | −0.42 | 0.2 | −0.14 | **0.86** |
| M_CC | 0.57 | −0.7 | 0.65 | −0.05 | 0.34 | −0.69 | 0.26 | −0.67 | **−0.9** | **−0.84** | 1 | −0.54 | 0.13 | −0.02 | 0.45 | **−0.99** |
| M_Alt | −0.51 | 0.25 | −0.47 | −0.07 | −0.25 | −0.2 | −0.78 | 0.39 | 0.27 | 0.11 | −0.54 | 1 | 0.18 | 0.27 | **−0.9** | 0.48 |
| M_Tap | −0.62 | −0.37 | −0.35 | −0.37 | 0.44 | −0.36 | −0.68 | −0.59 | −0.44 | −0.42 | 0.13 | 0.18 | 1 | −0.45 | −0.29 | −0.08 |
| M_Ori | −0.02 | −0.2 | −0.13 | 0.05 | **−0.87** | −0.33 | 0.15 | 0.54 | 0.25 | 0.2 | −0.02 | 0.27 | −0.45 | 1 | −0.39 | −0.01 |
| M_Slop | 0.61 | 0.11 | 0.72 | 0.45 | 0.46 | 0.25 | **0.82** | −0.47 | −0.24 | −0.14 | 0.45 | **−0.9** | −0.29 | −0.39 | 1 | −0.4 |
| SD_CC | −0.63 | 0.69 | −0.67 | 0.05 | −0.32 | 0.71 | −0.23 | 0.62 | **0.91** | **0.86** | **−0.99** | 0.48 | −0.08 | −0.01 | −0.4 | 1 |

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
