# Peer review of "Reduced-Impact Logging Maintain High Moss Diversity in Temperate Forests"

_forests, doi:10.3390/f12040383_

Round 1

Reviewer 1 Report

Overall, the paper provides some critical information on the effect of RIL harvesting on moss diversity. The experimental design is well thought out to answer the questions, although the sample size is a bit limited as indicated by the results.

While the reasoning and design of the paper is well thought out and presented there are some key issues that need to be addressed.

  • The paper needs a major edit for spelling and grammar. I have identified many areas where words and sentences need fixing but there are many more instances to be fixed.
  • There is a lack of detail in the methods of the paper and while it appears that the analysis answers the questions asked by the authors it is too difficult to determine if this has been done is an appropriate manner.

There are also a number of smaller issues detailed below.

Title

I think it should be maintains

Abstract

The abstract provides a good broad overview of the work although the authors could detail the specific key results a bit more.

L29: relationships not relations

L35-37: the logic of this sentence is unclear

Introduction

The introduction provides a god reasoning behind the purpose of this study and links in well with the local research.

L42: such as

L44-45: unclear sentence structure

L45 reduce

L46-48: sentence unclear

L50: six standard systems in Mexico or globally?

L54: “delimitated areas” I don’t understand this term

L53 – 55: I think you need a much bigger and better explanation of what is involved in RIL forestry for readers who are not familiar with this system

L76: moose???

Materials and Methods

This section needs a lot of work to make it clear to the readers what you have done. Specifically what data you used and when for each of your analyses and greater info on the reasoning why each analysis was chosen.

L92: constituted??? Not sure that is the right word

L93: such as

L110: wording of the last bit of this line is unclear

L114: When was the last time each of the managed sites had trees removed? And it is still unclear how RIL was undertaken at these sites, how many trees were removed per hectare??

L126: what area was the 10 quadrats based was it the same for each of the sites or different?

L146: Does the difference between when moss sampling was done (2016) and when the micro variables were recorded (2017) matter? There may have been changes in the forest structure over this time.

L146: where did you record the temp and light, you mention using tow recorders so they weren’t done at every quadrat. Also how did you measure daily temp was it the average which isn’t really a great descriptor of temp you would be better off sing the daily max or min or range.

Line 152: this sentence needs a reference

L154: remove besides and this sentence structure is also confusing.

L160: more detail on how you conducted the rank abundance curves

L159: what data did you do the NDMS on, site level or quadrat level??

L162: Wouldn’t it be good to include the rare species in your analysis as this is likely to be where the real differences between managed and control plots are, rare moss species are often rare because they need a specific set of conditions which may be removed by harvesting.

L164: Prior not previous, also need to explain the reasoning behind this transformation

L167: What data did you ue for the IndVal analysis quadrat or site level??

Line 176: What data did you use. If it was on site level data is that enough replication (n=6). Also how did you control for the effects of either forest type or site.

Results:
The results provide a relatively clear picture of the study, in some instances more detail is required.

L184: fifteen

L190: moss not mosses. Also 12 vs 18 is a bit of a difference, especially as sd is not that high so I don’t really agree with this sentence.

Line 203: is this for terrestrial mosses as well?

Line 212: I don’t think this is a range you are just reporting the numbers for two groups.

Line 225-231: This paragraph is difficult to read and needs a edit to clarify the information you are trying to get across

L232-234: These two sentences seem to contradict each other. Also, this section lacks detail, could explain the p values and stress values included in the figure?

L243; May need to add an explanation of specificity and fidelity for readers not aware of this type of analysis.

L245: LSV and RSV section could be a new paragraph

Discusion

While the discussion does a good job of providing a good level of supporting information from the literature, I think it fails to fully link the results from this study to the literature, especially in contrasting the lack of difference found here to other studies which have shown a difference. I also think there needs to be a greater acknowledgement of the limitations of this study such as the sample size as indicated by the R/E curves.

L259 Keep diversity where in managed patches or landscapes?

.L264-269: what are the major differences between harvesting using RIl and other systems that would lead to such a difference and do your result support that?

L273: what specific microclimate variable are you referring to?

L273-274: This sentence is confusing.

L278-282: Not sure what the point of these sentences are, are they species you found, or others found, how does it relate to your study?

L285: This would be a good spot to acknowledge the limited sampling size and the impact that it may have on your results.

L287: worthwhile mentioning that you removed these species for this study and therefore can’t make any conclusions about them.

L288-289: Species not in italics

L293: unique substrates such as ???

L291: rest of sites? All sites or just the managed ones?

L304-307: Sounds like D. sumichrastii has some downsides to being an indicator species but you pretty strongly recommend it in the end of this paragraph.

L319-322: Sentence is confusing and unclear

L326-328: Meaning of sentence is unclear

L339-340: These species were specifically not examined in this manuscript so you cannot make these conclusions.

L345: Qualification? Not sure that is the right word

Reviewer 2 Report

One of the most important indicators for RIL plots is the number of trees removed (possible by volume). The authors provided information about how the trees were selected for removal and the frequency of felling, but not the volume. This does not allow us to really assess the level of human impact. And this is the most important point that should be corrected.

l 42-43: "fragmentation" includes all others samples of influense, so this raw is not correct

l 49-51: Why "plantations" among different kinds of tree cutting?

l 76 "moose" instead of mosses

l 117 when using the term "prune", the authors refer to the removal of branches, not the whole tree. Is it true?

l 131 This is strange, because the direction can significantly affect the composition and coverage of epiphytic species

l 184 "Fiftheen" - mistake, it is better to use numbers as everywhere in the article

l 229-230 This sentence reads like that dominant species has a low abundance, please edit the sentence

l 288-289 species names should be in italics

Reviewer 3 Report

The manuscript “Reduced-impact logging maintain high moss diversity in temperate forests” provides interesting information on moss species richness and moss community between managed sites under RIL forest practices and conserved sites without any forest practice. This is a novel study for temperate forests in Mexico under RIL practices.

The effect of forest practices like RIL on different taxa is still under study. This is a controversial and challenging topic for the authors, so analyses and findings should be taken carefully. Based on this, the study needs more analyses to establish and support similarities in moss species richness and community. Analyses are suggested below.

The manuscript is easy to follow by the reader, but some suggestions are provided below to improve the understanding of the manuscript.

L19-20. This sentence should provide the value or importance of forest harvesting, otherwise, a solution to avoid biodiversity loss is to stop completely harvesting.

L31. Delete “statistically”.

L33. Considered instead of “regarded”.

L44. Affects biodiversity patterns actually, fix this accordingly.

L49-51. Present the six practices in some kind of impact intensity gradient on biodiversity (e.g. the least to the most harmful).

Use only one term, biodiversity or species richness, to be consistent throughout the manuscript.

L62-63. Unless the reference supporting this statement is a metanalysis or a literature review, the authors should provide at least more than two references supporting “low impact” for each of these groups.

The low impact of RIL practices on biodiversity is a controversial topic. The authors should not take this lightly and must provide examples of the high and low impact of RIL practices on fauna and vascular plants.

L66-68. Please provide direction to the variables (e.g. Low light? high humidity?) and so on for the other variables mentioned here.

L72. Not clear what “integrity” means.

L76-77. It may be re-written like “Thus, evaluating the impact of RIL on mosses is a crucial issue to ensure biodiversity preservation in the long-term.”

L80. Define site.

L82-83. Due to instead of “through”.

Mention the changes. Higher? Lower? Humidity

“ii” it’s not a hypothesis as it is written. Re-write this accordingly.

L91. Occurs.

L110. Dedicated instead of “destined”.

L143. What INEGI stands for?

Explain the “process” behind QGIS.

L185. Were exclusively terrestrial or epiphytic.

L190. For epiphytic mosses, right?

L191. Mention in the Table legend what both C and M stand for.

L226. Steep instead of “step”. What is the biological meaning of a steep slope from the analysis? Explain this here.

L229. Re-write this sentence for a better understanding. Could be like “There were a few dominant terrestrial moss species…

L245-246. Please refer back to LSV and RSV so the reader should not go back to find these acronyms.

It would be great to see how the LS and RS variables affect or influence the moss community in both managed and conserved sites through an NMDS analysis showing the variables as vectors.

This could be the same Figure 4 showing the LS ad RS variables as vectors.

Also, it seems necessary an analysis to compare LSV and RSV between managed and conserved sites. This will give a better idea of whether these sites are different or not.

“Of the LSV and RSV 245 analyzed, RSV did not show an impact on mosses diversity.”

The analyses to determine this should be included here.

L273. Well, it could be, but the authors should need an analysis to compare LSV and RSV to establish no differences between managed and conserved sites.

L278-282. These species should be shown in the Results, not here.

L287.Mention the specific requirements or suggest some based on the literature.

L288. Species names should be in italics.

L318. These.

L351. Understudied.

Comments to Authors

This manuscript “Reduced-impact logging maintain high moss diversity in temperate forests” provides interesting information on moss species richness and moss community between managed sites under RIL forest practices and conserved sites without any forest practice. This is a novel study for temperate forests in Mexico under RIL practices.

The effect of forest practices like RIL on different taxa is still under study. This is a controversial and challenging topic for the authors, so analyses and findings should be taken carefully. Based on this, the study needs more analyses to establish and support similarities in moss species richness and community. Analyses are suggested below.

The manuscript is easy to follow by the reader, but some suggestions are provided below to improve the understanding of the manuscript.

L19-20. This sentence should provide the value or importance of forest harvesting, otherwise, a solution to avoid biodiversity loss is to stop completely harvesting.

L31. Delete “statistically”.

L33. Considered instead of “regarded”.

L44. Affects biodiversity patterns actually, fix this accordingly.

L49-51. Present the six practices in some kind of impact intensity gradient on biodiversity (e.g. the least to the most harmful).

Use only one term, biodiversity or species richness, to be consistent throughout the manuscript.

L62-63. Unless the reference supporting this statement is a metanalysis or a literature review, the authors should provide at least more than two references supporting “low impact” for each of these groups.

The low impact of RIL practices on biodiversity is a controversial topic. The authors should not take this lightly and must provide examples of the high and low impact of RIL practices on fauna and vascular plants.

L66-68. Please provide direction to the variables (e.g. Low light? high humidity?) and so on for the other variables mentioned here.

L72. Not clear what “integrity” means.

L76-77. It may be re-written like “Thus, evaluating the impact of RIL on mosses is a crucial issue to ensure biodiversity preservation in the long-term.”

L80. Define site.

L82-83. Due to instead of “through”.

Mention the changes. Higher? Lower? Humidity

“ii” it’s not a hypothesis as it is written. Re-write this accordingly.

L91. Occurs.

L110. Dedicated instead of “destined”.

L143. What INEGI stands for?

Explain the “process” behind QGIS.

L185. Were exclusively terrestrial or epiphytic.

L190. For epiphytic mosses, right?

L191. Mention in the Table legend what both C and M stand for.

L226. Steep instead of “step”. What is the biological meaning of a steep slope from the analysis? Explain this here.

L229. Re-write this sentence for a better understanding. Could be like “There were a few dominant terrestrial moss species…

L245-246. Please refer back to LSV and RSV so the reader should not go back to find these acronyms.

It would be great to see how the LS and RS variables affect or influence the moss community in both managed and conserved sites through an NMDS analysis showing the variables as vectors.

This could be the same Figure 4 showing the LS ad RS variables as vectors.

Also, it seems necessary an analysis to compare LSV and RSV between managed and conserved sites. This will give a better idea of whether these sites are different or not.

“Of the LSV and RSV 245 analyzed, RSV did not show an impact on mosses diversity.”

The analyses to determine this should be included here.

L273. Well, it could be, but the authors should need an analysis to compare LSV and RSV to establish no differences between managed and conserved sites.

L278-282. These species should be shown in the Results, not here.

L287.Mention the specific requirements or suggest some based on the literature.

L288. Species names should be in italics.

L318. These.

L351. Understudied.

Round 2

Reviewer 1 Report

The authors have made some improvements to the manuscript but there are still many errors present throughout and a lack of clarity in the methods section

Title: should be maintains

L20: contributes.

L24: Sentence is confusing should it be “and its effectiveness has been assessed based on vascular plants”

L44 affects not affect

L53-55: Sentence is still unclear “sustainable forest management represents a series of management techniques that enable productive  practices and …….”

L56: I would remove this sentence completely it doesn’t add anything and I don’t think it is right.

L67 RIL management

L69: tree stands not three stands.

L75 mosses are associated to specific microenvironmental conditions

L120: including instead of as (both at start and end of line)

L135: conserved sites differed in size.

L137-146: This section is still very unclear and is very important for readers to understand the scale of the impact of RIL. In particular the wording makes it unclear how big these areas are is m1 107ha or 107 tree were removed per hectare? Are sites m2 and m3 much smaller.

L140 is particularly confusing does the disturbance last for 10 years and is this very different to M1? If so this probably needs to be discussed

I also still think it is very important to note how long since each site underwent disturbance. While you say that this information is available, I can’t work it out based on the current information. Especially as it appears as the sites vary in when they where last disturbed which will have significant impacts on your results.

L171: “The two recorders …”

L175: what is the spatial resolution of the INEGI data? Is it likely that each quadrat is on a different pixel of data? Also above you mention that this was done at the site level?

L200: coverture?? is this meant to be cover?

L202: how was the effect of site included in this analysis?

L209: you say you averages were calculated at both the site and quadrat level which of these data points went into the correlation analysis? If at the quadrat level how did you account for the effect of site?

L261: can probably remove the = signs

L275 epiphytic mosses

L276: both sites? Do you mean both forest types?

L281: significant p-value for terrestrial communities?

L282: can these values be presented in text

L284: I don’t understand what these references are here for. Might need more of an explanation of how they relate.

L284 moss not mosses

L286-288: This sentence doesn’t make sense. They are different but they aren’t?

Fig 3: seems very low resolution in the copy I have, may need checking

L333: From your results it looked as though RIL had a marginal impact on terrestrial communities. Same issue at L352

L346: Sentence is unclear maybe remove keeping

L349-351: This sentence sounds like you are reporting your results but then has reference. If comparing to another paper can you provide some extra detail if reporting your results remove the refs.

L359: Different reference style here.

Reviewer 2 Report

The authors responded to the comments and made appropriate corrections to the text of the manuscript, which now looks much better.

Author Response

Dear reviewer, we appreciate your comments. They were useful to improve our documentation. Our deep acknowledge.

Reviewer 3 Report

The Authors addressed properly the suggestions and comments from the reviewed version of the MS.

Author Response

(The authors gave the same response as above.)
